# Natural Appetite Control: Consumer Perception of Food-Based Appetite Regulating Aromas

**DOI:** 10.3390/nu15132996

**Published:** 2023-06-30

**Authors:** Jacek Łyczko, Michaela Godyla-Jabłoński, Natalia Pachura, Kinga Adamenko, Marta Klemens, Antoni Szumny

**Affiliations:** 1Department of Food Chemistry and Biocatalysis, Wrocław University of Environmental and Life Sciences, 50-375 Wrocław, Poland; natalia.pachura@upwr.edu.pl (N.P.); marta.klemens@upwr.edu.pl (M.K.); antoni.szumny@upwr.edu.pl (A.S.); 2Department of Human Nutrition, Wrocław University of Environmental and Life Sciences, 51-630 Wrocław, Poland; michaela.godyla@upwr.edu.pl; 3Department of Fermentation and Cereals Technology, Wrocław University of Environmental and Life Sciences, 51-630 Wrocław, Poland; kinga.adamenko@gmail.com

**Keywords:** HS-SPME Arrow, appetite regulation, food intake, health-care, elderly, obesity, sensory panel

## Abstract

According to the WHO, the number of overweight people (BMI ≥ 25) and obese people (BMI ≥ 30) is constantly growing. On the other hand, the number of elderly people (≥60 years old) in 2020 reached 1.4 billion worldwide. Both mentioned groups demonstrate their individual and characteristic appetite disorders. In light of the side effects of appetite stimulating drugs, which interfere with diabetics, hypertension and thrombosis medicines or diet supplements with doubtful effectiveness in reducing appetite, new and natural alternatives are highly demanded. Therefore, the present study focusses on the search for natural food aromas, which may have potential for appetite regulation. A survey was carried out among consumers with excess body weight (BMI ≥ 25) and the elderly (≥60 years old). Food products and meals pointed out by the survey participants were subjected to volatile analysis by HS-SPME Arrow followed by GC-MS. As a result, a group of volatiles and their odor characteristic were determined for appetite stimulation or reduction, which may suggest that the actual composition of food aroma is more significant than the character of the aroma. Those results may be a basis for designing appetite regulating agents, in which the mechanism of action will be based only on olfaction activity.

## 1. Introduction

In light of recent and up-coming changes in societies, health and demographical structure, appetite control is one of the crucial factors affecting the well-being. Unfortunately, according to the World Health Organisation (WHO), the numbers of overweight (BMI ≥ 25) and obese (BMI ≥ 30) individuals are constantly growing. In 2016, those numbers have reached 1.9 billion (38% of the worldwide population) and 650 million (13% of the worldwide population), respectively [1]. On the other hand, the number of elderly people (≥60 years old) in 2020 reached 1.4 billion [2]. Among this figure of elderly people, approximately 25–30% will be affected with anorexia of aging, which may be caused by appetite loss [3]. Therefore, both mentioned groups demonstrate their individual and characteristic appetite disorders.

Hunger and appetite are closely related terms, which are often confused. While hunger is an organism response for its homeostasis distraction (e.g., calories deficit), appetite, also named hedonic hunger, is a psychological need for food consumption, thus often a food craving stimulated by a constant food-fulfilled environment, even without an organism–calories deficient state [4,5]. The problem of cravings may be caused by a disturbed activity of AGPR neurones, which are responsible for signalling the calories needed, and what activity can be suppressed by food delivery. However, in addition, sensory stimulus related with food intake should be able to induce a similar suppression of neurone activity [6].

Sensory stimulants, which may reduce the activity of APGR neurones, are food, texture, taste, visual aspects and smell, among which, the last two could be used without actual food intake, which is crucial for both appetite stimulation and reduction [7,8]. Of course, the presence of food in front of the consumer has the predominant role in appetite changes, however, smell may be underrated regarding public health, in light of malnutrition or, in the contrary, the epidemic of overweight and obesity [9]. Recent studies on food odours, which are representative of high carbohydrate or protein or lipid content, showed that it is possible to stimulate an odour-specific (protein-related) appetite, however, overall food intake was not significantly affected [10]. Furthermore, it has been well documented that the odour of particular foods, such as sweet and savoury foods, may increase the appetite for food with a similar taste and/or calorie density, regardless of hunger levels [11]. Meanwhile, by in-depth food chemical analysis it was also determined that some food aromas, namely curry, nutmeg, or vanilla, due to the presence of particular chemical structures within their volatile organic compounds, may possess appetite-enhancing effects, which were examined in a mice model [12,13,14]. These chemical structures are mainly phenylpropanoid compounds, phenylpropanoid compounds with carbonyl group on the aliphatic chain, and compounds containing the vanillyl group.

Therefore, a study was designed to deeper explore the possible relations between the perception of food smell that enhances appetite or reduces appetite, and the actual volatile constituents of the food. For this purpose, the survey was carried out, dividing participants into two groups that both exhibit appetite dysfunctions—(i) overweight or obese individuals and (ii) elderly people (≥60 years old). Thereafter, the survey results were used to select a group of food products and meals, which were subjected ti the analysis of volatile organic compounds emitted from the food by the HS-SPME Arrow-GC/MS technique. Finally, 26 appetite regulating prototypes were designed and subjected to the sensory panel assessment.

## 2. Materials and Methods

### 2.1. Survey Participants

The research was carried out among consumers with excess body weight (Body Mass Index (BMI) ≥ 25) and the elderly (≥60 years old), based on an author’s questionnaire posted on the website via Google Forms or delivered via a paper version. The study consisted of 530 respondents aged 18 to 85 (including 329 women and 201 men). The questionnaire contained questions about the basic data differentiating the respondents, such as age, current body weight and height, place of residence, type of employment, and education. More than half of the respondents were people aged 60 and older (*n* = 274; 51.7%). The most numerous group among the respondents was that of people with secondary education (*n* = 186; 35% of the respondents), and the least numerous was that of primary education (*n* = 33; 6% of the respondents). A total of 31.5% of the respondents declared higher education (*n* = 167). Most often, they lived in rural areas (*n* = 217 people; 41%) and cities with up to 50,000 inhabitants (*n* = 147; 28%). The majority of the respondents lived in the Opolskie Voivodeship (*n* = 228; 43%) and Dolnośląskie (*n* = 120; 23%), and the least lived in in the Zachodniopomorskie Voivodeship (*n* = 5; 0.9%) or were temporarily or permanently outside Poland (*n* = 3; 0.6%). Based on the declared values of body weight and height of the examined person, the BMI [kg·m^−2^] was calculated. The average BMI value among the respondents was 29.13 kg·m^−2^ (defined as overweight, pre-obesity), whereas the lowest recorded value was 14.68 kg·m^−2^ (underweight), and the highest was 58.82 kg·m^−2^ (obesity class III).

### 2.2. Survey Procedure

The first part of the questionnaire enclosed a certificate to be completed with data on gender and age, education, place of residence, and financial situation of the respondent. To assess the appetite of consumers and the risk of weight loss, in the second part of the questionnaire, the Simplified Nutritional Appetite Questionnaire (SNAQ) (questions 1–4) was used. Lastly, in the third, most substantial part, the survey consisted of 49 questions, among which 47 were closed multiple-choice questions and 2 were open-answer questions).

Questions 5–17 provided an example of a food product (e.g., tea with sugar, chilli pepper) and were intended to evaluate the influence of food consumption on participant feelings, including appetising, satisfying, or hunger stimulating. Questions 18–33 provided an example of food product aroma (e.g., cheese, bread) and were intended to evaluate the influence on participants’ further food choices. Questions 34–51 were intended to evaluate which of the provided food products or their aromas might evoke a particular sense of safety or satiety or which one’s were appetising or unappetising or provided a blissful feeling as an emotional response. Food products that were selected for questions 5–51 were due to a few factors. The main aim was to include foods from various groups including meat, fruits, vegetables, diary, whole meals, sweets, etc.; moreover, the listed food products had to be common for the population, which was an object of the study, to avoid the situation in which some products will be unknown for participants. The last 52–53 questions were intended to evaluate the emotional response of individual participants with regard to examples of satisfying or appetising smells. The survey may be found as Appendix A.

### 2.3. Chemical Analysis

The screening of what is deemed potentially useful for appetite control, chemical constituents of food products and meals was performed by the headspace solid-phase microextraction Arrow (HS-SPME Arrow), followed by GC-MS analysis. Briefly, the representative amount of food products and meals (full list is available as Appendix A) was placed into a 20 mL headspace vial along with internal standard (undecanon-2-one Sigma-Aldrich, Steinheim, Germany) and then was pre-conditioned for 10 min at a temperature adequate to the generally accepted food regular consumption temperature, as per health and safety standards, and subjected to extraction for 30 min with 1.10 mm DVB/C-WR/PDMS SPME Arrow fiber (Shimadzu, Kyoto, Japan). Thereafter, analytes were desorbed at 250 °C with a split ratio of 1:5 and separated on a ZB-5MSi (30 m × 0.25 mm × 0.25 μm) column (Phenomenex, Torrance, CA, USA). The GC operational conditions: carrier gas (helium) flow 1.00 mL·min^−1^; 50 °C kept for 2 min, then to 180 °C at a rate of 3.0 °C·min^−1^, then to 270 °C at a rate of 20.0 °C·min^−1^ and kept for 5 min. MS operational conditions: ion source temperature 220 °C; interface temperature 250 °C, scanning mode 40–400 *m*/*z*.

The identification of volatiles found in the food aroma profile was performed by a comparison of experimentally obtained linear retention indices (LRIs), calculated against the C_7_-C_40_ n-alkane mix (Sigma-Aldrich, Steinheim, Germany), and experimentally obtained mass spectra, with those available in the Flavour and Fragrance Natural and Synthetic Compounds GCMS Library (FFNSC 3) (Shimadzu Company, Kyoto, Japan) or NIST 17 Mass Spectral and Retention Index Libraries (NIST20) (The National Institute of Standards and Technology, Gaithersburg, MD, USA). As potential targets, only compounds with a mass spectrum similarity score ≥90% and LRI ± 15 were considered. For samples LID_44–LID_57 and LID_60, the list of volatile compounds were determined on the basis of AroChemBase (Alpha MOS, Toulouse, France).

### 2.4. The Design of Appetite Regulating Prototypes

On the basis of the combined results of the survey and chemical analyses, the appetite regulating prototypes were designed. For this purpose, various food applicable aromas (SUPER AROMAS, Sobucky Poland sp. Z o.o. Sobucky Ltd., Sp.k., Lublin, Poland), spices (OLMIX s.c., 43–340 Kozy, Poland), essential oils (Herbiness Inez Rogozińska, Chomiec, Poland) and pure volatile compounds (Sigma-Aldrich, Steinheim, Germany) were purchased. The prototypes were designed by mixing the food applicable flavourings with selected pure, volatile organic compounds and diluting them with ≥99% triethyl citrate (Sigma-Aldrich, Steinheim, Germany). The prepared compositions were subjected to sensory panel assessment. The list of prepared appetite regulating prototypes was provided as Appendix A.

### 2.5. Sensory Analysis

Sensory analysis of the appetite regulating prototypes was carried out by a sensory panel consisting of research and teaching staff from the Wroclaw University of Life Sciences. The selection of panellists was based on their experience in sciences related to human nutrition and food, and nutrition technology. Selection also took into account sensory indisposition or limitations in odour recognition. A total of 10 panellists, 6 women and 4 men, were selected, and were between the ages of 28–51 years. 

The panellists, in individual booths with controlled temperature, humidity and light conditions, received the coded samples in a random order. Each sample was assessed according to the prepared questioner (Appendix A). Due to the large number (26) of appetite-regulating prototypes, their evaluation was divided into 5 sessions, held on separate days at the same time.

### 2.6. Statistical Analysis

In order to examine the dependence between discrete variables, multiway tables and the Chi-square maximum likelihood test (NW) were used. The study also investigated the dependence between discrete variables using nonparametric Spearman correlations. The correlation coefficients assumed values in the range of <−1; 1>. For the data obtained during chemical analysis, STATISTICA 13.3 software for Windows (StatSoft, Krakow, Poland) was used. The hierarchical cluster analysis (HCA) with Ward’s linkage and Euclidean distance was applied, and the strict (33%) Sneath’s criterium was used to highlight the sensory evaluation results.

## 3. Results

### 3.1. Survey

The abnormal SNAQ result (score ≤ 14 points, which means a risk of further weight loss in 6 months) was found in 21.5% of the study group, including 73 women and 41 men. Taking into account the age of the respondents, the risk of weight loss was observed in 39 people aged 18–59 years and in 75 people aged ≥60 years old. A very high positive correlation (0.76) was found between BMI [kg/m^2^] and the perception of body shape among the respondents, therefore, BMI and the declared sex assigned at birth were assumed as a grouping (independent) variable. A positive, however, weak correlation (0.13), between the BMI value and the sum of points obtained in the sNaQ questionnaire, was also demonstrated.

There were statistically significant differences in appetite sensations after theoretical exposure to vanilla-flavoured products. In the case of respondents of normal weight, 41 people found that this smell did not affect appetite, whereas 76 did. Among 237 overweight and obese people, this smell did not affect food consumption, and 162 respondents reported such an effect. Almost 1/3 of the respondents (*n* = 146; 27.5%) would consume vanilla pudding after exposure to the aroma of vanilla, and 22% (*n* = 117) would consume vanilla ice cream. Furthermore, there were statistically significant differences in appetite sensations after exposure to the smell of bananas. As many as 68% of the respondents (*n* = 361 people) declared their willingness to eat any of the proposed products, including up to 264 overweight or obese people. More than half of the respondents (*n* = 280; 52.8%) declared their willingness to eat bananas after being exposed to their aroma, while nearly 14% of the group (*n* = 74) would be willing to eat milk chocolate. Only 8 people (1.5%) said they would like to eat crackers, 10 people (1.9%) wanted fruit jam, and 12 (2.2%) said they would eat salty snacks. Less than 20% of the respondents (*n* = 94) have declared a reduction of appetite for sweets after exposure to the smell of green lettuce. Among the proposed dishes in the questionnaire, the most pleasing for the respondents were the dumplings, while the tomato soup with noodles turned out to be the least pleasing, while among the fruits, the most pleasing for the respondents were strawberries, apples and pears, and the least was lemons and pineapples. According to the respondents, the most appetizing smell was the smell of coffee (*n* = 301; 56.8%) and vanilla (*n* = 158; 29.8%), and the least appetizing was the smell of maple syrup (*n* = 200; 37.7%) and rosemary (*n* = 157; 29.6%). There was also a weak positive correlation (0.11) between BMI value and hunger suppression after drinking tea with sugar. The selected survey results were presented in Table 1.

What is interesting is that there were statistically significant differences in the sense of safety after contact with various types of smells. We stated negative correlations between the BMI value and the sense of security after the smell of boiled egg and oatmeal in milk/porridge (−0.25 and −0.23, respectively). A weak and positive correlation (0.13) was observed between BMI and the smell of vanilla ice cream. Slightly different results were investigated for women (we asserted/found an average, negative correlation between BMI and the sense of security after exposure to the smell of oatmeal in milk). Almost 45% of the respondents (*n* = 237; 44.7%) felt safe after exposure to the smell of coffee. The sense of bliss among 225 respondents (42.5%) was caused by the smell of coffee, in 159 people (30%) by the smell of vanilla ice cream, and in 144 people (27.2%) by the smell of chocolate ice cream. According to the respondents, the most filling fruits were bananas (*n* = 351; 66.2%), apples (*n* = 203; 38.3%) and pears (*n* = 150; 28.3%). The least filling were watermelons (*n* = 320; 60.4%) and lemons (*n* = 214; 40.4%). The most filling smells were those of meat products: grilled sausage (*n* = 298; 56.2%), naturally smoked bacon (*n* = 157, 29.6%) and roast beef (*n* = 150; 28.3%). The smell of black tea, kefir (a type of fermented milk drink with a sour taste) and rosemary turned out to be the least filling for 42.9%, 34.1% and 25.1%, respectively, of the study group (the detailed statistical analysis is provided as Appendix A).

### 3.2. Screening of Chemical Compounds with Appetite Control Potential

The survey conducted has delivered the broad spectrum of food products and meals (full list is available as Appendix A) that were described as ones with the potential to stimulate or a reduce appetite. The detailed chemical analysis by the HS-SPME Arrow technique has revealed that despite the various characteristics of food products and meals pointed out by survey participants, the volatiles emitted from them have presented a consistency. This consistency was confirmed by HCA (Figure 1), which has shown that volatiles may be divided into two groups, which refers to products identified as ones with the potential to stimulate or reduce appetite, respectively.

### 3.3. Sensory Analysis of Appetite Regulating Prototypes

The sensory analysis of appetite regulating prototypes, in order to select those to be qualified for further stages of the project’s research, i.e., consumer studies, was carried out. The summary presented in Table 2 shows that the panellists differentiated the prototypes in terms of their potential to stimulate (12 prototypes) or reduce (4 prototypes) appetite. Furthermore, 10 prototypes were not categorized as either having the potential to stimulate or reduce appetite.

## 4. Discussion

The volatile organic compounds present in food products and meals aromas may have a significant influence on appetite, which was well proven by the research by Zoon et al. (2016) [11], in which they have determined that food odours may stimulate the appetite for food products and meals with similar aroma and/or energy density. On the other hand, McCrickerd & Forde (2016) [8], in their broad review, have summarised that short-term ambient food odours may influence food choices, which should be useful for the appetite down-regulation for those deemed undesirable in diet food products and meals. Such a strategy corresponds with the findings by Zoon et al. (2016) [11] regarding the relationship between particular aromas and the desire to eat similarly smelling foods. The present research focused on the investigation of the consumers’ opinion on potential food products and meals and their aromas, which can contribute to the stimulation of appetite, or, conversely, the reduction. The survey participants provided multiple responses describing different food products and meals to reduce or stimulate appetite. However, on the chemical level of the aroma profiles of food products and meals, the differences were reduced, which was proved as pointed out in the survey; food products and meals for appetite stimulation are characterised by a convergent group of volatiles and the same phenomenon was observed for food products and meals and their aromas identified by participants as ones with the potential to reduction appetite. In light of the study by Ogawa et al. [12,13,14], it may mean that the presence of actual chemical structures is more important than the aroma character which they create. The aforementioned studies proves that the presence of specific structures, such as phenylpropanoid structure, phenylpropanoid structure with carbonyl group on the aliphatic chain and vanillyl group, may be linked to the capacity of volatiles to stimulate the appetite. Nonetheless, the present study was not focused on the investigation regarding the chemical structure issue.

In the work of Yin et al. (2017) [7], they suggest that it may not be possible to influence appetite only by aroma or taste, but it is necessary to use both simultaneously; however, in their research, they used the strawberry aroma, which as we determined by the survey, is not an aroma adequate to the reduction of appetite. Similar research was performed by Morquecho-Campos et al. (2020) [10] and Morquecho-Campos et al. (2021) [15], although improving food intake by conscious or non-conscious exposition for odour stimuli representing particular macronutrients (proteins, carbohydrates, and fat), respectively, also showed that this research resulted with the conclusion that smell stimulation by food-related odours do not have an impact on food intake. However, they used particular aromas (butter, cream, corn, bread, duck, chicken, and cucumber) which may be more considered meal ingredients than food consumed independently, while the present research highlighted food products and meals, which may be consumed as individual meals or snacks.

The volatiles identified in the present study were divided into two groups for both the stimulation and reduction of appetite, respectively. The very first group (Figure 1A) consists of organic acids (C_6_, C_8_ and C_9_), linear aldehydes (C_7_, C_8_, C_10_, C_12_), limonene, furfural, α-pinene, benzaldehyde and 6-methyl-5-hepten-2-one, and are described as cheesy, fatty aromas (organic acids), citrus and fresh (limonene, linear aldehydes, 6-methyl-5-hepten-2-one) or woody, camphorous, bakery, bitter odours (furfural, α-pinene, benzaldehyde) [16]. Such odours may be linked to the association with acidity or fresh bread [17,18], which has a positive influence on saliva production, thus the preparation of the organism for food intake. On the other hand, volatiles with the potential to reduce appetite (Figure 1B) create a more complex group; the included volatiles represent fruity (e.g., isoamyl acetate), floral (e.g., α-terpineol, linalool), herbal (e.g., β-pinene, camphor, carvone), balsamic (e.g., 2-acetylfuran) or aldehydic, fatty (e.g., linear aldehydes, linear alcohols) odours [16].

On the basis of this finding, we propose that it is possible to create the aroma compositions which may be used for moderate appetite regulation, in which, according to our best knowledge, will be the very first attempt for that strategy. Therefore, we have chosen the overall food products and meals aromas, among those pointed out by survey participant, to be used as a base for appetite regulating products and fortified them with particular volatile organic compounds, which we found to be the most often occurring in the volatile organic compounds profiles. The sensory panel analysis has shown that different compositions regarding appetite regulating prototypes may induce different responses. In most cases, the character of a particular odour had the potential to stimulate appetite, which may be related to the current type of human–food relationship. Lowe and Butryn (2007) [4] have introduced considering the term ‘hedonic hunger’, which refers to easily stimulated appetite, which prevails over the hunger.

## 5. Conclusions

Food aroma perception is an individual matter. Nevertheless, despite the differences in survey respondents’ responses, it appears that the actual volatile composition is a crucial factor. On the basis of surveys linked with chemical analysis, there is a possibility of highlighting chemical compounds or their groups, which may have an application in appetite regulation. For appetite stimulation compounds, such as organic acids (C_6_, C_8_ and C_9_), linear aldehydes (C_7_, C_8_, C_10_, C_12_), limonene, furfural, α-pinene, benzaldehyde, and 6-methyl-5-hepten-2-one, they were determined as the ones with the highest potential. For appetite reduction compounds with odor characteristics such as fruity (e.g., isoamyl acetate), floral (e.g., α-terpineol, linalool), herbal (e.g., β-pinene, camphor, carvone), balsamic (e.g., 2-acetylfuran) or aldehydic, fatty (e.g., linear aldehydes, linear alcohols), they were determined to be those with the highest potential. On the basis of chemical analyses and the consumer survey, 26 appetite regulating prototypes were designed and assessed by the sensory panel. As a result, 12 were evaluated as the ones with the potential to stimulate appetite, 4 were evaluated as the ones with the potential to reduce appetite, and 10 were evaluated as the ones with no particular potential. 

## 6. Patents

On the base of this study patent applications, designated as: P.445245; P.445246; P.445247; P.445248; P.445251 and P.445252 have been submitted to Polish Patent Office.

## Figures and Tables

**Figure 1 nutrients-15-02996-f001:**
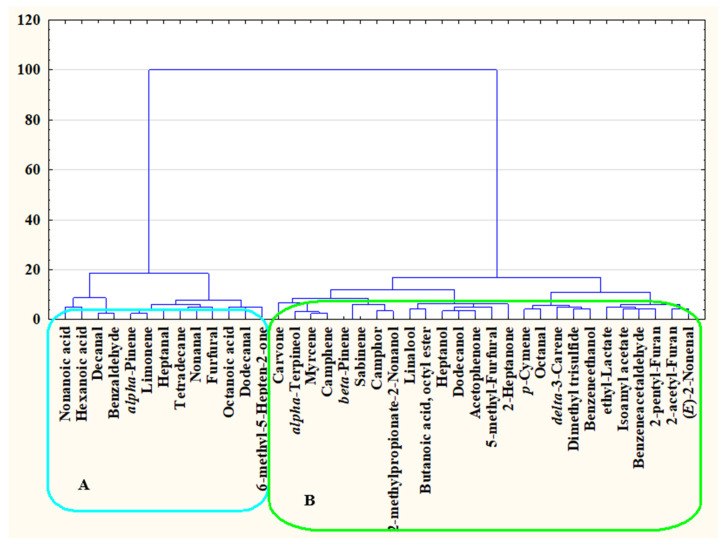
Hierarchical cluster analysis (HCA) of volatiles present in various types of food products and meals in relation to appetite regulation; (**A**)—volatiles linked to FMPs pointed out as ones with the appetite stimulating potential; (**B**)—volatiles linked to FMPs pointed out as ones with the appetite reducing potential.

**Table 1 nutrients-15-02996-t001:** The influence of food products and meals and their aromas on the consumer’s feelings.

Name	Source	Effect	na ^a^	Classification
Apples	Product	The most appetizing	127	Appetite stimulation/Appetite reduction
		The most satiating fruits	203	
Banana	Smell	Would like to eat bananas	280	Appetite stimulation/Appetite reduction
		Would like to eat milk chocolate	74	
	Product	The most satiating fruits	351	
Black tea	Smell	The least satiating smells	108	Appetite stimulation
Chocolate ice cream	Smell	The sense of bliss	144	Appetite reduction
Coffee	Smell	The sense of safe	237	Appetite stimulation (black coffee)/Appetite reduction (milk coffee)
		The sense of bliss	225	
		The most appetizing	301	
Dumplings	Product	The most appetizing	147	Appetite stimulation
Green lettuce	Smell	Reduction appetite for sweets	237	Appetite stimulation
Grilled sausage	Smell	The most satiating smells	298	Appetite reduction
Kefir	Smell	The least satiating smells	130	Appetite stimulation
Lemons	Product	The least appetizing	268	Appetite stimulation
		The least satiating fruits	214	
Maple syrup	Smell	The least appetizing	200	Appetite stimulation
Naturally smoked bacon	Smell	The most satiating smells	157	Appetite reduction
Pears	Product	The most appetizing	112	Appetite stimulation
		The most satiating fruits	150	
Pineapples	Product	The least appetizing	186	Appetite reduction
Roast beef	Smell	The most satiating smells	150	Appetite reduction
Rosemary	Smell	The least appetizing	157	Appetite stimulation
		The least satiating smells	131	
Strawberries	Product	The most appetizing	311	Appetite stimulation
Vanilla	Smell	Would like to consume vanilla pudding	146	Appetite stimulation
		Would like to consume vanilla ice cream	117	
		The most appetizing	158	
Vanilla ice cream	Smell	The sense of bliss	159	Appetite reduction
Vegetable salad	Product	The least appetizing	128	Appetite reduction
Watermelons	Product	The least satiating fruits	320	Appetite stimulation

^a^ number of respondents.

**Table 2 nutrients-15-02996-t002:** Summary of appetite regulating prototypes sensory panel assessment.

Sample	Panel Qualification	Intensity of the Effect	Odour Strength	Aroma Character
AP_27/C612	appetite stimulation	perceptible/intense	intense	fruity; wild berries; watermelon; nice
AP_11/N217	appetite stimulation	perceptible/intense	intense	fruity; banana; lollipop; jelly
AP_12/Z094	appetite stimulation	perceptible	intense	sweet; marshmallow;
AP_15/A501	appetite stimulation	perceptible	subtle/perceptible	fruity; wild berries; melon
AP_16/A048	none	-	perceptible	fresh; lime; menthol
AP_17/Y603	appetite stimulation	perceptible	perceptible	fruity; apples; bananas; sweets
AP_13d/C853	none	-	intense/perceptible	coffee; coffee candy; caramel
AP_15d/B303	none	-	intense/perceptible	fruity; citrus; cocktail
AP_16d/D418	none	-	perceptible/intense	citrus; lime; household chemicals; refreshing
AP_17d/W078	appetite stimulation	perceptible	perceptible/intense	sweet; candy; fruity
AP_18d/G371	none	-	intense	fruity; citrus; peach; dough
AP_21/X034	appetite reduction	perceptible	intense	dessert; chocolate; pudding; candy; biscuits
AP_22/K083	appetite stimulation	intense	intense	fruity; sweet; vanilla; apple
AP_23/K359	none	-	subtle	cocoa; chocolate; biscuits; nuts
AP_24/S005	none	-	intense	minty; irritating; air-refresher
AP_25/I581	none	-	perceptible	fruity; apple; quince;
AP_26/M408	none	-	perceptible/very intense	chemical; irritating; medicinal; smoked
AP_0/T042	appetite stimulation	subtle/perceptible	intense	sweet; cake; pudding
AP_0 */V681	appetite stimulation	perceptible	intense	sweet; sweet cream; chocolate; candy
AP_5/R789	appetite reduction	perceptible	perceptible	unpleasant; chemical; solvent-like
AP_6/O301	appetite stimulation	perceptible	intense	woody; smoked; resin
AP_7/A681	appetite stimulation	perceptible	subtle	woody; smoked; heavy
AP_8/Q461	none	-	subtle	woody; trees
AP_4 ***/P986	appetite stimulation	perceptible/subtle	perceptible	wood; resin; vanilla; earthy
AP_28/H090	appetite reduction	subtle	perceptible	fruity; apricot; green fruits; hops
AP_31/F699	appetite reduction	perceptible/intensive	perceptible	herbs; grass; green; citrus air-refresher

* and *** were used to distinguish the samples AP_0 and AP_4 from samples AP_0* and AP_4***; this marking was introduced due to similar composition of appetite regulating prototypes

## Data Availability

Data is available at the open science framework website https://osf.io/5zpn2/?view_only=4dd6bac728274925b39c5f8ef388ef2b (archived on 29 June 2023)).

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
