# Peer review of "Natural Appetite Control: Consumer Perception of Food-Based Appetite Regulating Aromas"

_nutrients, 2023, doi:10.3390/nu15132996_

Round 1

Reviewer 1 Report

This article presents a captivating discussion on the impact of distinct aromas on appetite, rendering it highly publishable. However, there are certain areas that require scrutiny. Although extensive research has been conducted on the effects of food aroma on consumer perception, the author has provided insufficient logical elaboration in the preface. It would be more appropriate to give an introduction that encompasses the entire spectrum of food aromas and their effects on consumers. Additionally, a singular variable addition experiment would greatly enhance the author's findings. Consider, for instance, how consumers' preferences for a favored food are affected when a specific compound is added to it.

This article presents a captivating discussion on the impact of distinct aromas on appetite, rendering it highly publishable. However, there are certain areas that require scrutiny. Although extensive research has been conducted on the effects of food aroma on consumer perception, the author has provided insufficient logical elaboration in the preface. It would be more appropriate to give an introduction that encompasses the entire spectrum of food aromas and their effects on consumers. Additionally, a singular variable addition experiment would greatly enhance the author's findings. Consider, for instance, how consumers' preferences for a favored food are affected when a specific compound is added to it.

Author Response

Dear Reviewer,

thank you for your kind words and your valuable remarks. We hope that our corrections will find your acceptance. Our detailed response is given in pdf file.

1) Although extensive research has been conducted on the effects of food aroma on consumer perception, the author has provided insufficient logical elaboration in the preface. It would be more appropriate to give an introduction that encompasses the entire spectrum of food aromas and their effects on consumers. Thank you for this suggestion. We have improve the Introduction and deliver the overview of food-related aromas in light of appetite level. Please see lines 62-70.

2) Additionally, a singular variable addition experiment would greatly enhance the author's findings. Consider, for instance, how consumers' preferences for a favored food are affected when a specific compound is added to it. This is a very interesting point of view. Actually, within the project, the part presented in the manuscript study, we plan experiments with the participation of consumers. During those studies, consumers will a) receive some volatile compositions and after the smelling will be asked to fill out the questionnaire regarding their appetite and b) will receive some volatile compositions and after the smelling will be asked to eat a meal (we will check the amount of leftovers). Nevertheless, this is the further part of the project, while we do not want to hold back with the publication of our first results due to pretty high interest in such topics among other scientific groups. We hope that our planes and point of view will be acceptable to you.

Kind regards, 

Jacek Łyczko

Reviewer 2 Report

The study was designed to explore the possible relationships between food smell perception in increasing or reducing appetite and volatile food constituents.

Below are my considerations:

In the abstract put a conclusion highlighting the relevance of the results found.

I would like to clarify if the research was evaluated by an Ethics Committee, since it is research involving human beings.

Regarding the group of elderly people (> 60 years old), can you generalize that all these people present some appetite alteration?

What were the criteria for choosing the food products listed in the questionnaire. Clarification.

Does the scientific literature support the claims that the smells of the foods selected can affect appetite? Wouldn't this be subjective?

Figure 1 should come before item 4 Discussion.

Will the same person always have the same reaction when exposed to a certain aroma or is it related to the moment. How to consider these variations.

The studies mentioned in the discussion are also based on consumer opinion and collected by forms. Discuss.

The work should focus more on the volatile composition, since the perception of aromas is variable.

Minor editing of English language required

Author Response

Dear Reviewer,

thank you for your kind words and your valuable remarks. We hope that our corrections will find your acceptance. Our detailed response is given in pdf file.

1) In the abstract put a conclusion highlighting the relevance of the results found. The Abstract has been corrected. Please see lines 15-30.

2) I would like to clarify if the research was evaluated by an Ethics Committee, since it is research involving human beings. As we stated at the end of the manuscript we did not receive the evaluation of particular Ethics Committee, since such was not existing at the Wrocław University of Environmental and Life Sciences at the moment of study performance (2021). The Ethics Committee at UPWr was established in January 2023. Nevertheless, since this study is a part of bigger project financed by the National Center of Research and Development (Poland), during the application the detailed description of particular studies and experiments was described within the application forms. The substantial part, including survey study present in this study, was evaluated in to steps – 1 st by 3 independent Reviewers and then 2 nd by the Selection Committee of Lider XI program consisting of scientist from different fields, including ethical and medicinal sciences. At that stage, no concerns were reported regarding the planned survey study. We have clarified this issue in the manuscript. Please see lines 349-355.

3) Regarding the group of elderly people (> 60 years old), can you generalize that all these people present some appetite alteration? Thank you for this remark. It was not our intention to generalize that all elderly people present some appetite alteration. Our hypotheses were based on that, this is a group with significant risk of appetite alteration presence. This assumption is based on the phenomena of ‘anorexia of aging’, which may affect approximately 25-30% of elderly (Jadczak and Visvanathan 2019; DOI 10.1007/s12603-019-1159-0). One of the factors, which may cause this condition is loss of appetite. We have clarified this issue in the manuscript. Please see lines 42-44.

4) What were the criteria for choosing the food products listed in the questionnaire. Clarification. The food products selection for questionnaire were chosen due to few considerations. The main goal was to include foods from various groups including meat, fruits, vegetables, diary, whole meals, sweets etc; moreover, the listed food products had to be common for the population which was an object of the study, to avoid the situation, in which some products will be unknown for participants. For this purpose one of manuscript co-authors (Mrs. Michaela Godyla-Jabłoński) was invited to cooperation due to her experience and knowledge regarding human nutrition and diets. We have clarified this issue in the manuscript. Please see lines 113-117.

5) Does the scientific literature support the claims that the smells of the foods selected can affect appetite? Wouldn't this be subjective? We agree that the selected smells may be subjectively received by consumers. According to our findings within the present study, we assume that the actual composition of the aroma is more important than the odour character; thus it is important that the smell stimuli would have a pleasant aroma, however, the presence of particular compounds will be more important than the odour character to stimulate or reduce appetite. Due to this fact, for the further studies within this project, we have chosen the strategy to use pointed out food aromas as a base and fortified them with compounds we found in the highest number within the samples during volatile analysis. We have added some part regarding this issue to the manuscript. Please, see lines 146-155 and 301-312.

6) Figure 1 should come before item 4 Discussion. The Figure 1 has been placed before Discussion section.

7) Will the same person always have the same reaction when exposed to a certain aroma or is it related to the moment. How to consider these variations. Thank you for this valuable observation. We fully agree that the perception of overall aroma may differ due to multiple factors. Furthermore, our findings support this observation, since consumers respond to the same question in the survey (e.g. “Is there any smell after which you would feel hungry? If yes, then what kind?”) gave various answers. However, the chemical analysis of various food products and meals that were pointed out have revealed that on the chemical level, the volatile organic compounds that create particular aromas were similar. This may suggest that the perception of aroma is less important than the compounds present in it. We discuss the results of other studies, which indicates that the particular chemical structures have a significant influence on appetite enhancing. Please see lines 268-274.

8) The studies mentioned in the discussion are also based on consumer opinion and collected by forms. Discuss. Thank you for this valuable insight. That’s true that the studies which we referred to were mainly based on hedonically studies with consumers. Therefore we supplemented the Discussion with some studies, which are focused more on the methods which do not consider the consumers opinion, but were conducted with animal models. Please see lines 268-274.

9) The work should focus more on the volatile composition, since the perception of aromas is variable. We have added the broad part regarding the possibility of use of our findings in purpose to compose some appetite regulating prototypes, which will action will only be based on olfaction activity. Please see lines 146-167, 242- 249, 301-312 and 324-328. Also, due to broad spectrum of food products and meals subjected to volatile chemical analyses, the results of those analyses (volatile composition of 60 objects) was given as Supplementary materials 2; in this file all data, including volatiles identification factors, is presented. We hope that this solutions will be acceptable and will exhaust the answer to this remark.

Kind regards, 

Jacek Łyczko
